# ASYMMETRIC EMBEDDING MODELS FOR HIERARCHICAL RETRIEVAL: PROVABLE CONSTRUCTIONS AND A PRETRAIN-FINETUNE RECIPE

## ABSTRACT

Dual encoder (DE) models, where a pair of matching query and document are embedded into similar vector representations, are widely used in information retrieval due to their efficiency and scalability. However, DEs are known to have a limited expressive power due to the Euclidean geometry of the embedding space, which may compromise their quality. This paper investigate such limitations in the context of *hierarchical retrieval*, the task where the document set has a hierarchical structure and the matching documents for a query are all of its ancestor nodes. We first prove the feasibility of representing hierarchical structures within the Euclidean embedding space by providing a constructive algorithm for generating effective embeddings from a given hierarchy. Then we study the problem of learning such embeddings when the hierarchy is unknown, which is often the case in practice where only samples of matching query and document pairs are available during training. Our experiments reveal a "lost in the long distance" phenomenon, where retrieval accuracy degrades for documents further away in the hierarchy. To address this, we introduce a pretrain-finetune approach that significantly improves long-distance retrieval without sacrificing performance on closer documents. Finally, we validate our findings on a realistic hierarchy from Word-Net, demonstrating the effectiveness of our approach in retrieving documents at various levels of abstraction.

## 1 INTRODUCTION

*Information retrieval* (Mitra et al., 2018) systems aim to find the most relevant documents within a large database in response to a user query. A common example is an internet search engine that, for instance, sifts through all available products and returns those that are the most relevant to a customer's search terms. Dual encoder (**DE**) is one of the most commonly used models for information retrieval due to their simplicity and scalability. DEs function by encoding both queries and documents by vector representations, often using deep neural networks (Guo et al., 2016; Chang et al., 2020; Zhao et al., 2024). Then, similarity between a query and a document is calculated using their Euclidean distance or inner product, enabling scalable retrieval through approximate nearest neighbor search (Johnson et al., 2019; Guo et al., 2020; Chern et al., 2022).

This paper focuses on a specific case of information retrieval where the underlying database exhibits a hierarchical structure. In particular, we consider databases with an organization that can be represented as a directed acyclic graph (DAG), where each node corresponds to a document and each edge represents a directional relationship between a pair of documents (see Figure 1). This hierarchical organization, prevalent in many real-world systems (Ravasz & Barabási, 2003; Adcock et al., 2013), leads to the challenge of *hierarchical retrieval*. In this context, relevant responses to a query include not only the "same meaning" document but also its ancestors in the hierarchy. For instance, consider the task of keyword targeting where advertisers build a keyword list for their products so that their ads surface if a user query is relevant to keywords in the list. A particular way of doing this[1] is that an advertiser puts a general keyword, e.g., "Sport shoes", which is to be retrieved not

---

[1] https://support.google.com/google-ads/answer/11586965?hl=en#:~:text=A%20keyword%20match%20type%20that,specific%20form%20of%20the%20meaning

only when the user query means the same as this particular keyword, but also when the query is more specific, e.g., "Kid's running shoes" (see Figure 1). Meanwhile, user queries that are not considered more specific, e.g. "Kid's shoes" for the keyword "Sport shoes", should not retrieve the keyword.

DEs is a tempting choices for solving hierarchical retrieval tasks due to their widespread use for retrieval. However, DEs have known limitations in representation capacity due to the Euclidean geometry of the embedding space (Menon et al., 2022). Such limitations may be particularly problematic for the hierarchical retrieval tasks. For instance, to effectively retrieve both "Kid's shoes" and "Running shoes" for a query "Kid's running shoes" (see Figure 1), a DE would need to place the document embeddings of these two parent documents in close proximity to each other as they need to be both close to the embedding of the query. However, this proximity can lead to unintended consequences. If a user searches for "Running shoes", the model might incorrectly retrieve "Kid's shoes" which is irrelevant to the query. One approach to address this shortcoming is to consider alternative measures of similarities beyond Euclidean distance, such as the hyperbolic geometry (Nickel & Kiela, 2017; Tifrea et al., 2018; Ganea et al., 2018). Another approach in the literature is to use region instead of point based embeddings, such as Order Embeddings (Vendrov et al., 2015) and Box embeddings (Vilnis et al., 2018; Chheda et al., 2021). While these methods demonstrate superior performance in handling hierarchical structures, they are no longer amenable to nearest neighbor search hence not suited for retrieval applications with a large set of documents.

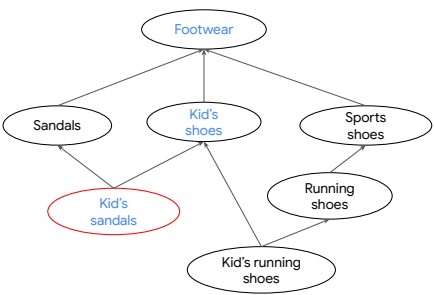

Figure 1: Illustration of the hierarchical retrieval task. Documents form a hierarchy. Given a query, e.g. "Kid's sandals" (circled in red), the task is to retrieve all its ancester nodes in the hierarchy (shown in blue).

**Contributions.** This paper explores DEs for hierarchical retrieval, aiming at understanding their capabilities and limitations and developing practical algorithms for improving their quality.

Firstly, given the constraints of the Euclidean geometry and the asymmetric nature of hierachical retrieval, it is unclear whether it is even possible to accurately represent a hierarchical similarity structure via Euclidean embeddings. Moreover, even if such a representation does exist, a priori it may require very large dimensionality, possibly scaling with the number of points in the dataset. In Section 3, we resolve this uncertainty with an optimistic answer to the question. Specifically, we present a constructive algorithm which, given a hierarchical structure, generates a set of embeddings in $\mathbb{R}^M$ which solve the hierarchical retrieval task on that structure. Moreover, the dimension of the embeddings is at most $O(h \log n)$, where $n$ is the number of points in the dataset and $h$ is the depth of the hierarchy.

The constructive algorithm in Section 3 requires precise knowledge of the document hierarchy. In practice, this hierarchy is often unknown and the embeddings need to be learned from a training dataset composed of matching query and document pairs. Hence, it is unclear if learning on such data is able to find good representations even though they exist. Section 4 explores this learning process by experimenting with a synthetic tree-structured hierarchy. By controlled experiments with varying the depth and width of the tree, we show that the learning process not only finds good representations but can achieve this with dimensions much less than those required in the constructive algorithm, for a wide range of hierarchies.

Continuing on the study of learned embeddings, in Section 5 we provide a fine-grained analysis that reveals a "lost in the long distance" phenomenon, which severely compromise their quality. This refers to the behavior that documents with longer distances from the query becomes increasingly more difficult to retrieve. Towards mitigating the issue, we first demonstrate the inadequacy of naively upsampling of such pairs, which exhibits a quality tradeoff between short distance and long distance pairs. We then present a *pretrain-finetune* recipe, where a DE is first pretrained on a regular data and then finetuned on a dataset focusing solely on pairs with queries and their long-distance ancestors. Finally, this approach is validated in Section 6 on a realistic graph from the WordNet, a large lexical database of English. On this dataset, we show that given a specific query e.g. "cat", the

learned embeddings using our pretrain-finetune approach can retrieve documents that are at different levels of abstraction from the query, e.g., "cat", "feline", "carnivore", "placental", etc.

## 2 Hierarchical Retrieval and Dual Encoders

Let $\mathcal{Q} = \{q_i\}_{i=1}^n$ and $\mathcal{D} = \{d_j\}_{j=1}^m$ be a collection of $n$ queries and $m$ documents, respectively. For each $i \in \{1, \ldots, n\}$, let $S(q_i) \subseteq \mathcal{D}$ be a set that contains documents that are the most relevant to the query $q_i$. The goal of information retrieval is to return a ranked list of the documents in $\mathcal{D}$ for each $q_i$, with the top ones being those in $S(q_i)$. In this section, we formally define *hierarchical retrieval* as a particular case of this information retrieval problem. We then discuss dual encoders (DEs) and set a roadmap on studying their effectiveness for hierarchical retrieval.

### 2.1 Hierarchical retrieval

In hierarchical retrieval, we assume that the document set $\mathcal{D}$ has a hierarchical structure described by a directed acyclic graph (DAG), which recall is a directed graph with no directed cycles. Let $\mathcal{G} = (\mathcal{V}, \mathcal{E})$ be a DAG associated with the set of document $\mathcal{D}$, where $\mathcal{V} = \mathcal{D}$, i.e. each vertex corresponds to a document, and $\mathcal{E} = \mathcal{D} \times \mathcal{D}$ are directional edges between pairs of documents. For hierarchical retrieval we assume the following: if any document $d_j \in \mathcal{D}$ is defined as relevant for a given query $q_i$, i.e., $d_j \in S(q_i)$, then any document $d_{j'} \in \mathcal{D}$ for which there is a path in $\mathcal{G}$ that goes from $d_j$ to $d_{j'}$ is also considered relevant to $q_i$, i.e., $d_{j'} \in S(q_i)$.

### 2.2 Dual encoders

DEs are asymmetric embedding models composed of a query encoder $f_q(\cdot, \boldsymbol{\theta}_q) : \mathcal{Q} \to \mathbb{R}^M$ and a document encoder $f_d(\cdot, \boldsymbol{\theta}_d) : \mathcal{D} \to \mathbb{R}^M$, which map all the queries and documents into the same embedding space $\mathbb{R}^M$. Here, $\boldsymbol{\theta}_q$ and $\boldsymbol{\theta}_d$ are parameters of DE that can be learned from a training dataset. Once learned, the top-$k$ matching documents for a query $q_i \in \mathcal{Q}$ can be obtained by computing the inner product score between its query embedding $f_q(q_i, \boldsymbol{\theta}_q)$ and all the document embeddings, i.e., $\{f_d(d_j, \boldsymbol{\theta}_d)\}_{j=1}^M$, followed by filtering out those with the $k$ largest scores.

When training a DE, we assume the availability of a dataset containing pairs of matching query and document pairs. Let $\{(i_k, j_k)\}_{k=1}^B \subseteq \{1, \ldots, m\} \times \{1, \ldots, n\}$ be a batch of such data, where $i_k$ and $j_k$ are indices of the query and document, respectively. Then, a DE may be trained by minimizing the following *batch softmax loss*:

$$L(\boldsymbol{\theta}_q, \boldsymbol{\theta}_d) = \frac{1}{B} \sum_{k=1}^B \text{cross-entropy}\left( \text{softmax}\left( \frac{\boldsymbol{D}(\boldsymbol{\theta}_d)^\top \cdot f_q(q_{i_k}, \boldsymbol{\theta}_q)}{T} \right), \mathbf{1}_k \right),$$
$$\text{where } \boldsymbol{D}(\boldsymbol{\theta}_d) = [f_d(d_{j_1}, \boldsymbol{\theta}_d), \ldots, f_d(d_{j_B}, \boldsymbol{\theta}_d)] \in \mathbb{R}^{M \times B}. \quad (1)$$

In above, $\mathbf{1}_k \in \mathbb{R}^B$ is a vector with the $k$-th entry being 1 and all other entries being 0, and $T$ is a hyper-parameter that we fix to be 20 unless specified otherwise. This loss comes from casting the retrieval as a classification task, where each document in the batch is treated as a class and the class label for the query $i_k$ is $k$.

In the rest of the paper, we study the effectiveness of DE trained with minimizing the loss in Eq. (1) for hierarchical retrieval. This problem is challenging as the solution to Eq. (1) depends on multiple factors, including the structure of the hierarchy, the architecture of the encoder models, and the optimization process, etc. Hence, we consider simplifications that makes the analysis tractable:

- In Section 3, we focus on the purely geometric question: Given the hierarchy $\mathcal{G}$, does there exist a collection of query and document embeddings that solves the hierarchical retrieval problem? This is equivalent to making a strong assumption that we can freely put the query and document embeddings anywhere, irrespective of the learning process, and that the networks $f_q$ and $f_d$ are sufficiently expressive to produce any collection of embeddings.

- In Section 4, we study learned embeddings from minimizing the following loss function

$$L(\{\boldsymbol{q}_i\}_{i=1}^n, \{\boldsymbol{k}_i\}_{i=1}^m) = \frac{1}{B} \sum_{k=1}^B \text{cross-entropy}\left( \text{softmax}\left( \frac{[\boldsymbol{d}_{j_1}, \ldots, \boldsymbol{d}_{j_B}]^\top \cdot \boldsymbol{q}_{i_k})}{T} \right), \mathbf{1}_k \right), \quad (2)$$

That is, we consider the query and document embeddings, i.e., $\{\boldsymbol{q}_i\}_{i=1}^n \subseteq \mathbb{R}^M$ and $\{\boldsymbol{k}_i\}_{i=1}^m \subseteq \mathbb{R}^M$, as free optimization variables that do not depend on the network parameters $\boldsymbol{\theta}_q$ and $\boldsymbol{\theta}_d$. Equivalently, this is assuming that the networks $f_q$ and $f_d$ are implemented as lookup tables with one embedding associated with each $q \in \mathcal{Q}$ and $d \in \mathcal{D}$, respectively. Such a simplification enables us to focus on the effect of the hierarchy on the properties of the learned embeddings, without dealing with a complicated deep network.

## 3 THE GEOMETRY OF HIERARCHICAL RETRIEVAL

In this section, we present an algorithm for constructing a collection of query and document embeddings that solves the hierarchical retrieval task. In fact, our construction will work for a much more general task, where the document structure may not be hierarchical.

Consider a query set $\mathcal{Q} = \{q_i\}_{i=1}^n$ and a document set $\mathcal{D} = \{d_j\}_{j=1}^m$. For each $i \in \{1, \ldots, n\}$, let $S(q_i) \subseteq \mathcal{D}$ be a set that contains documents that are relevant to the query $q_i$. We assume that $|S(q_i)| \leq s$ for all $i \in [n]$. Our goal is to construct $M$-dimensional embeddings for the $n$ queries and $m$ documents such that every query is close to each of its relevant documents, and not similar to any other document. Formally, the objective is to find unit vectors $\boldsymbol{q}_1, \ldots, \boldsymbol{q}_n, \boldsymbol{d}_1, \ldots, \boldsymbol{d}_m \in \mathbb{R}^M$, along with a threshold $r \in [-1, 1]$, such that for all $i \in [n]$ and $j \in [m]$ the following holds:

- **Case 1:** If $j \in S(q_i)$, then $\langle \boldsymbol{q}_i, \boldsymbol{d}_j \rangle \geq r + \epsilon$.
- **Case 2:** If $j \notin S(q_i)$, then $\langle \boldsymbol{q}_i, \boldsymbol{d}_j \rangle < r - \epsilon$

**A constructive algorithm.** Our constructive algorithm is the following.

---

**Algorithm 1** A constructive algorithm for hierarchical retrieval

---

**Input:** Query set $\mathcal{Q} = \{q_i\}_{i=1}^n$, document set $\mathcal{D} = \{d_j\}_{j=1}^m$, and $\{S(q_i) \subseteq \mathcal{D}\}_{i=1}^m$

  1: Take $\{\boldsymbol{d}_j = \frac{\widehat{\boldsymbol{d}}_j}{\|\widehat{\boldsymbol{d}}_j\|_2}\}_{j=1}^n$, where $\{\widehat{\boldsymbol{d}}_j\}_{j=1}^n \subseteq \mathbb{R}^M$ are drawn i.i.d. from standard Gaussian.

  2: For each $i$, take $\boldsymbol{q}_i = \frac{\widehat{\boldsymbol{q}}_i}{\|\widehat{\boldsymbol{q}}_i\|_2}$, where $\widehat{\boldsymbol{q}}_i = \sum_{j \in S(q_i)} \widehat{\boldsymbol{d}}_j$

**Output:** Query emebeddings $\{\boldsymbol{q}_i\}$ and document embeddings $\{\boldsymbol{d}_j\}$.

---

In words, the algorithm simply involves taking the document embeddings as random Gaussian vectors, then setting each query embedding as the sum of the document embeddings of its relevant documents. Then, both embeddings are normalized to have unit $\ell_2$ norm.

**Correctness of the constructive algorithm.** We will parameterize our bounds in terms of the quantities $n, \epsilon, s$. For simplicity, we will assume that $n = m$ in our analysis. Note that this can be achieved WLOG by padding the smaller of the two quantities with dummy queries (or documents). The goal is to minimize the dimension $M$.

Our main theoretical result is the following which shows that such embeddings can be created with dimension linear in the parameter $s$. Observe that for the case of the hierarchical retrieval problem, $s$ is at most the number of ancestors of a given node $i$ in the graph. If the DAG is in fact a tree, then $s$ *is at most the height of the tree.*

**Theorem 1.** *For any input sets $S(q_1), \ldots, S(q_n) \subset [n]$ satisfying $|S(q_i)| \leq s$ for all $i \in [n]$, one can construct vectors solving the dual-encoder embedding construction task with parameter $\epsilon$ in dimension $M = O(\max\{s \log n, \frac{1}{\epsilon^2} \log n\})$.*

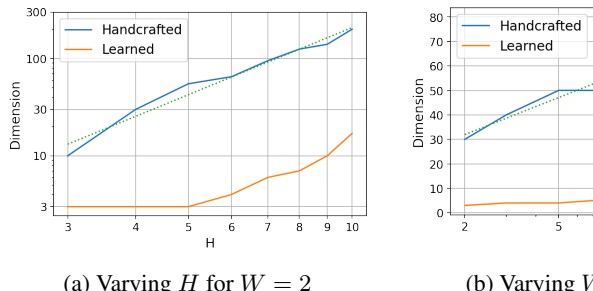

(a) Varying $H$ for $W = 2$          (b) Varying $W$ for $H = 4$

Figure 2: Comparison of learned DE and handcrafted DE for hierarchical retrieval with a tree-structure of degree $W$ and height $H$. For *Handcrafted*, we generate embeddings according to Algorithm 1. For *Learned*, we learn embeddings by minimizing the loss in Eq. (1) using randomly sampled data of a sufficient amount until convergence. In both cases, we experiment with an increasing sequence of embedding dimensions $M$ until the retrieval is successful, and report that dimension on the y-axis. Successful retrieval means that the averaged recall@k is above $95\%$ for each of 5 independent trials on a test set of $10k$ matching pairs, where $k$ is the number of relevant documents for each test query. Dotted lines are least squares fittings of the *Handcrafted*. The left figure uses log-log scale and the right figure uses log scale on x-axis.

## 4   Learning a Dual Encoder for Hierarchical Retrieval

While the construction algorithm in Section 3 provides a solution that solves the hierarchical retrieval problem with provable guarantees, it requires knowing the hierarchical structure a priori. In this section, we study embeddings learned from minimizing the objective in Eq. (2), aiming at understanding if it is possible to learn such good embeddings in practice.

**Experimental setup.** Towards that, we consider a particular kind of hierarchical structures represented by perfect $W$-trees, where each node has $W$ child nodes and all leaf nodes of the tree are at the same level. A perfect $W$-tree is described by two parameters, namely $W$ which we refer to as the *degree*, and $H$ which is the number of levels and we refer to as the *height*.

For the purpose of hierarchical retrieval, we take the query set $\mathcal{Q}$ and the document set $\mathcal{D}$ to both be the set of nodes of a $W$-tree (excluding the root). For each query $q \in \mathcal{Q}$, we assume that its matching document set $S(q) \subseteq \mathcal{D}$ contains $d = q$, as well as all the ancestors of $d$. In other words, the tree is interpreted as encoding a hierarchy of the documents with edges pointing from each child node to its parent node.

We use the following procedure to sample training and evaluation datasets, unless specified otherwise. First, a query is sampled by drawing a node with equal probabilities from all nodes of the tree. Then, we sample a node with equal probabilities from the set of documents that match $q$. Training is conducted by optimizing Eq. (2) with gradient descent. For evaluation we use recall@$k$, i.e., the percentage of $(q, d)$ pairs in the evaluation set for which $d$ is one of the $k$ documents that have the largest inner product score with $q$, with $k$ being the total number of relevant documents for $q$.

**Learned vs constructed embeddings.** Recall that Algorithm 1 provides a construction of query and document embeddings which solve the hierarchical retrieval if the dimension of the embedding space satisfies $M = O(s \log n)$. For a tree-structured hierarchy that we consider here, $s$ and $n$ in this formula are given by $s = H$ and $n = O(W^{H-1})$. Hence a $M = O(H^2 \log W)$ is needed for successful retrieval by this construction. This section compares this with that of the learned embeddings, by learning embeddings on trees with varying $H$ and $W$.

In Figure 2, we report the dimension of the embedding space (i.e., $M$) that is needed to achieve successful hierarchical retrieval, as a function of $H$ on the left and $W$ on the right. First, the curves for handcrafted embeddings align well with the theoretical result that $M = O(H^2 \log W)$ is sufficient. For example, in Figure 2a we perform a line fitting in the log-log space and obtain a slope of 2.29, whereas from $M = O(H^2 \log W)$ we can derive a slope of 2. In Figure 2b we perform a line fitting

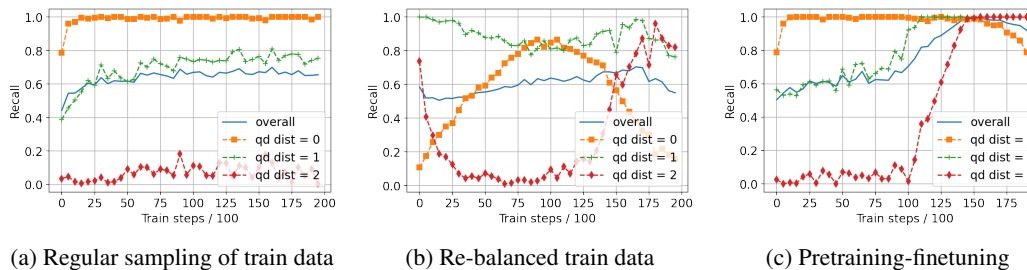

(a) Regular sampling of train data    (b) Re-balanced train data    (c) Pretraining-finetuning

Figure 3: Recall on test sets with varying distances $\in \{0, 1, 2\}$ between query and document (i.e., *qd dist* in the legend), for embeddings of $M = 3$ dimensions learned on a tree of $H = 4$ and $W = 5$. (a) For embeddings trained on data sampled by a default sampling procedure, the recall for $(q, d)$ pairs with a distance of 1 or 2 are low. (b) By training on re-balanced data, where $(q, d)$ pairs with a distance of 1 or 2 are up-sampled, the recall for pairs with a distance of 1 and 2 are significantly improved at the cost of a drastic decrease in recall for pairs with distance 0. (c) With a pretraining-finetuning procedure described in Section 5.2, recalls at all distances are close to 100% near 15000 steps. In particular, the overall recall (i.e., weighted average of recall at all 3 distances) improves to 97%, compared to 66% in (a) and 70% in (b).

in the space of $\log(W)$ and obtains a slope of 16.5, which aligns well with a slope of $H^2 = 16$ obtained from $M = O(H^2 \log W)$.

In comparing handcrafted with learned embeddings, we see from Figure 2 that the latter obtains successful retrieval with a much smaller $M$. This result demonstrates that learning is able to find embeddings that are even better than the handcrafted ones.

# 5   IMPROVING HIERARCHICAL RETRIEVAL VIA PRETRAIN-FINETUNE

Here we provide a fine-grained analysis of the embeddings learned on tree-structured hierarchies that are considered in Section 4 and identify a "lost in the long distance" phenomenon. We then present a recipe for addressing this problem which leads to better quality embeddings.

## 5.1   LOST IN THE LONG DISTANCE

We provide an analysis of the retrieval quality by evaluating for pairs at varying distances. Here, the distance between a matching $(q, d)$ pairs is defined as the difference between the level of the node $q$ and the level of the node $d$. For example, a distance of 0 means that $q$ and $d$ correspond to the same node, and a distance of 1 means that $d$ corresponds to the parent node of $q$.

Towards that, we consider a tree of $H = 4$ and $W = 5$, and fix an embedding dimension of $M = 3$. In this case, for a query at level 4, the matching document could be from levels 2, 3, or 4 which correspond to a distance of 2, 1, and 0, respectively (recall that the root node does not correspond to any query or document). Likewise, queries at level 3 have matching documents with distance 0 and 1, and queries at level 2 have matching documents of distance 0 only. To evaluate recall at different distances, we sample an evaluation dataset and split it to three parts, which contain pairs with distance 0, 1, and 2, respectively. We then calculate recall on these three parts separately and report the results in Figure 3a. It can be seen that the learned embeddings can almost retrieve perfectly for pairs with a distance 0. However, the embeddings do not work well for distance 1, and perform poorly for distance 2. In other words, learned embeddings do not work well for pairs where the document is distant away from the query.

A straightforward idea for addressing this problem is to re-balance the training dataset, so that it contains more pairs of longer distances. To test this idea, we consider the following two sampling strategies:

- *Regular* sampling. This refers to the sampling procedure described in Section 4. With $H = 4$ and $W = 5$, it can be calculated that pairs with distances 0, 1, and 2 are sampled with probability 38%, 35%, and 27%, respectively.

- *Heavy-Tail* sampling. This refers to a sampling strategy where pairs with distances 0, 1, and 2 are sampled with probability 0%, 50%, and 50%, respectively.

We then create a re-balanced dataset by mixing data from regular and heavy-tail sampling above with a ratio of $p : 1 - p$. Figure 3b shows results with $p = 0.03$. It can be observed that recalls for pairs with distance 1 and 2 are significantly improved and reaches a level of beyond 80% towards the end of training. However, we also observe that this comes at the cost of a significantly reduced recall on pairs with distance 0. Finally, we tune $p$ with varying values in $\{0.01, 0.1, 0.3\}$ and report the results in Figure 6 (see Appendix). The results confirm that re-balancing training data cannot effectively solve the issue of lost in long distance.

### 5.2 A PRETRAIN-FINETUNE RECIPE

We present a pretrain-finetune recipe to address the issue of lost in the long distance. This recipe simply requires first doing a pretraining on data from the regular sampling, then finetune the embeddings on the heavy-tail distribution using a much smaller learning rate (we used $0.05\times$). Here, regular and heavy-tail distributions refer to those described in Section 5.1.

The results of this recipe is reported in Figure 3c, where the pretraining stage and the finetuning stage take place in the first and second 10,000 training steps, respectively. We observe that in the finetuning stage, the retrieval quality for pairs with distance 1 and 2 quickly picks up and reach a level of close to 100%. After that, the model exhibits an overfitting since the quality for pairs with distance 0 starts to decrease. This overfitting may be expected since the finetuning stage does not have any training data with distance 0. In practice, one may apply an early stopping by monitoring recall on a heldout set, which will enable us to obtain good quality embeddings for hierarchical retrieval. In particular, the peaking overall recall (i.e., averaged over pairs of all distances) reaches 97% which is a huge improvement over using the default data sampling (which gives 66% recall) and over using re-balanced training data (which gives 70% at peak performance near 17000 steps).

## 6 EXPERIMENTS ON REAL DATA

In this section, we demonstrate the effectiveness of the pretrain-finetune recipe in Section 5 on data with a realistic hierarchical structure. Specifically, we use WordNet (Miller, 1995), a large lexical database of English where the nouns, verbs, adjectives, and adverbs are grouped into *synsets* that represent synonyms. The set of synsets is equipped a binary *hypernym* relation, e.g., "chair" is the hypernym of "armchair". This relation may be described by a DAG with nodes corresponding to synsets and edges pointing from a synset to another one that is its hypernym.

For our experiments, we use the 82,115 noun synsets from WordNet. Then, the query and document sets, i.e., $\mathcal{Q}$ and $\mathcal{D}$, are taken to be those synsets. For each query $q \in \mathcal{Q}$, the matching documents $S(q)$ are synsets for which there is an edge that points from $q$ to it in the graph of the transitive closure of the DAG associated with the hypernym relation. For example, the set of matching documents for the query "cat" includes "cat", "feline", "carnivore", "placental", etc. In practice, we make a slight modification to this definition by restricting to $(q, d)$ pairs with a distance of at most 8, so that the learning task is easier. Here, distance between two synsets is defined as the length of the shortest path that connects them in the hypernym DAG.

**Lost in the long distance.** As a baseline, we assume that there is a *regular* sampling procedure which we can use for generating matching pairs from WordNet. This procedure is described as follows. First, a query $q$ is sampled uniformly at random from all 82,115 synsets. Then, a document is sampled uniformly at random from the set of all matching documents to $q$. In our experiment, we sample 10M such pairs as the training dataset, and optimize the objective in Eq. (2) using SGD with a learning rate of $0.5$, momentum of $0.9$, batch size of 4096, for 50k iterations (i.e., 20 epochs).

To evaluate the quality of the learned embeddings, we use a test set of size 10k generated in the same way, and report recall@k as the quality metric, where $k$ is the total number of relevant document for a query. In particular, we report recall@k averaged over all samples in the test set, reported as the column "Overall" in Table 1, and slices with different distances (i.e., 0, 1, ..., 8) between query and document. Finally, we pick the checkpoint that achieves the best overall recall on a validation set of size 10k as the final output.

Table 1: Quality of learned embeddings for hierarchical retrieval on WordNet. *Regular sampling* refers to first sampling a query then a document uniformly at random from the set of all matching documents. *Rebalanced* means a mixture of data from regular sampling with a proportion $p$ and a heavy-tail data of proportion $1-p$ where long-distance pairs are upsampled. *Pretrain-finetune* is the method of first pretraining on regular sampling data then finetuning on heavy-tail data. Quality is measured by averaged recall@k on a test set. **Our method (i.e., pretrain-finetune) enables good retrieval quality for query-document pairs at all distances.**

| Method | Query-document distance | | | | | | | | | Min | Overall |
|---|---|---|---|---|---|---|---|---|---|---|---|
| | 0 | 1 | 2 | 3 | 4 | 5 | 6 | 7 | 8 | | |
| *Embedding dimension = 16:* | | | | | | | | | | | |
| Regular sampling | 100.0 | 62.8 | 46.6 | 33.9 | 20.9 | 11.9 | 7.2 | 2.9 | 1.0 | 1.0 | 43.0 |
| Pretrain-finetune | 100.0 | 57.1 | 46.4 | 47.9 | 50.2 | 53.6 | 53.1 | 47.3 | 32.0 | 32.0 | 60.1 |
| *Embedding dimension = 32:* | | | | | | | | | | | |
| Regular sampling | 100.0 | 90.8 | 79.2 | 62.3 | 46.4 | 31.8 | 20.1 | 14.8 | 8.4 | 8.4 | 61.8 |
| Pretrain-finetune | 100.0 | 77.3 | 76.5 | 80.4 | 83.5 | 84.2 | 84.3 | 80.1 | 67.3 | 67.3 | 87.3 |
| *Embedding dimension = 64:* | | | | | | | | | | | |
| Regular sampling | 100.0 | 93.9 | 86.9 | 76.8 | 60.2 | 46.8 | 36.1 | 28.8 | 19.4 | 19.4 | 71.4 |
| Rebalanced (p=0.01) | 0.6 | 46.9 | 69.7 | 67.2 | 56.3 | 44.7 | 39.4 | 34.9 | 36.6 | 0.6 | 41.8 |
| Rebalanced (p=0.03) | 2.8 | 49.6 | 69.4 | 64.2 | 52.1 | 43.2 | 37.7 | 31.8 | 32.7 | 2.8 | 40.7 |
| Pretrain-finetune | 100.0 | 90.8 | 91.6 | 92.7 | 92.6 | 91.8 | 90.9 | 87.3 | 75.7 | 75.7 | 92.3 |

Table 2: Retrieved synsets for selected queries with 64-dimensional embeddings. For each of the three queries considered here, we list the relevant documents in the row "Groundtruth" with an ascending order in their distance to the query. For "Regular sampling" and "Pretrain-finetune", we list retrieved top-$k$ documents in an ascending order of $k$. Undescored documents are those retrieved but not in the groundtruth.

| *Query = "cat"* | | | | | | | | | |
|---|---|---|---|---|---|---|---|---|---|
| Groundtruth | cat | feline | carnivore | placental | mammal | vertebrate | chordate | animal | organism |
| Regular sampling | cat | feline | carnivore | placental | mammal | wildcat | domestic cat | vertebrate | canine |
| Pretrain-finetune | cat | feline | chordate | vertebrate | animal | placental | mammal | carnivore | wildcat |
| *Query = "recliner"* | | | | | | | | | |
| Groundtruth | recliner | armchair | chair | seat | furniture | furnishing | instrumentality | artifact | whole |
| Regular sampling | recliner | armchair | seat | chair | furnishing | furniture | article | ware | toy dog |
| Pretrain-finetune | recliner | armchair | seat | chair | furnishing | furniture | instrumentality | artifact | cleaning pad |
| *Query = "motorist"* | | | | | | | | | |
| Groundtruth | motorist | driver | operator | causal agent | physical entity | | | | |
| Regular sampling | motorist | operator | driver | floridian | foe | | | | |
| Pretrain-finetune | motorist | operator | driver | physical entity | causal agent | | | | |

In Table 1, we report the results with a varying dimension of embedding space in $\{16, 32, 64\}$. In all the cases we observe that quality degrades rapidly with the distance between query and document.

**Rebalanced data sampling is insufficient.** The lost-in-the-long distance phenomenon may be explained by the distribution of the regular sampling training data, which is heavily biased towards query-document pairs with very short distances (see Figure 4). To explore the effect of data distribution, we assume the availability of a Heavy-Tail training dataset generated as follows. First, a query $q$ is sampled in the same way as in the normal sampling. Then, the matching document for $q$ is sampled with a probability *proportional to the distance between the document and* $q$ (instead of uniform sampling). In our experiment, we sample 1M such heavy-tail pairs. Then, we create a *rebalanced* dataset where each batch has $p \times 4096$ pairs from regular sampling data and $(1 - p) \times 4096$ from heavy-tailed data. Results with $p = 0.01$ or $p = 0.03$ for embedding dimension 64 are reported in Table 1. It can be seen that rebalanced data improves the

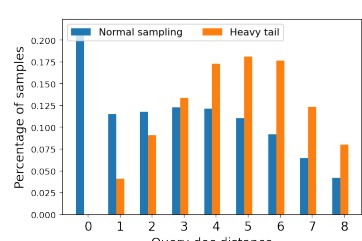

Figure 4: Distribution of regular sampling and heavy-tail data over varying query-document distances.

retrieval quality on long distance pairs but at a large cost of hurting quality on short distance pairs.

Table 3: Spearman correlation score on Hyperlex for embeddings trained on WordNet with the normal sampling procedure (i.e. "Normal sampling") vs embeddings further finetuned on heavy-tail data (i.e., "Pretrain-finetune"). We used 5-dimensional embeddings to be consistent with prior work. Among comparing methods, the results for Order Embedding and WN-Basic are taken from Vulić et al. (2017). WN-Euclidean represents 5-dimensional embeddings trained on WordNet using Euclidean distance and the result is taken from Nickel & Kiela (2017).

| Method | OrderEmbed | WN-Basic | WN-Euclidean | Regular sampling | Pretrain-finetune |
|---|---|---|---|---|---|
| $\rho$ | 0.195 | 0.240 | 0.389 | 0.350 | 0.415 |

**Pretrain-finetune offers a solution.** We evaluate the pretrain-finetune method, where the embeddings are first pretrained on the normal sampling data for $50k$ iterations, then finetuned on the heavy-tail dataset *only*, i.e, not mixed with normal sampling data, for $20k$ more iterations. During finetuning, the only changes we make are to reduce learning rate to 1,000 smaller and to use a temperature of 500 in lieu of 20 (see Eq. (2)). The results, presented in Table 1, demonstrate a significant retrieval quality improvement for long distance document, leading to a much higher overall recall. In Table 2, we further provide a visualization of the retrieved documents for three selected queries. These examples show that normal sampling tends to miss the long distance pairs and the pretrain-finetune recipe is effective in fixing many of such errors.

**Hypernymy evaluation.** We further evaluate the learned embeddings on HyperLex (Vulić et al., 2017), a dataset for evaluating how well a model captures the hyponymy-hypernymy relation between concept pairs. Towards that, we train 5-dimensional embeddings on the nouns of WordNet using pairs generated from regular sampling as before, and report the result in Table 3 under the entry "Regular sampling". This method is similar to WN-Euclidean reported in Nickel & Kiela (2017), which also trains 5-dimension embeddings on WordNet using the Euclidean distance, and hence the quality of these two entries are similar. Finally, we test our pretrain-finetune recipe by doing a finetuning the embeddings trained on regular sampling data using the heavy tail data. The result shows a large improvement over regular sampling and also surpasses the previous result of WN-Euclidean.

**Comparison with hyperbolic embeddings.** Document embedding using a hyperbolic space is a popular approach for addressing the shortcomings of embedding with the Euclidean space and have demonstrated successes for hierarchical structures. As we explain in the related work, hyperbolic embeddings cannot solve the hierarchical retrieval problem as the embeddings are not amenable to approximate k nearest neighbor search algorithms. Here we further provide empirical evidence that hyperbolic embeddings are also insufficient in their retrieval capabilities.

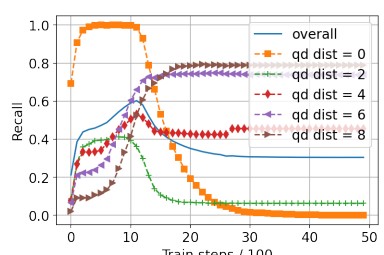

Figure 5: Retrieval quality of hyperbolic embeddings at varying query-document distances.

Towards that, we train $64$ dimensional hyperbolic embeddings on the same 10M normal sampling data as before and report the recall for query-document pairs at different distances in Figure 5[2]. It can be seen that hyperbolic embeddings struggle to obtain a good balance between pairs with short vs long distances. Specifically, the model first learns to retrieve pairs with short distances, i.e. with distance 0 and 2. As it starts to be able to retrieve longer distance pairs, the quality on the short distance pairs begin to drop very quickly.

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

# A    APPENDIX

## A.1    PROOF OF THEOREM 1

We now present the proof of our construction of embeddings that solves the dual encoder embedding construction task. For notation, in what follows, we write $a = x \pm y$ to indicate the containment $a \in [x - y, x + y]$

*Proof of Theorem 1.* We begin by drawing random Gaussian vectors $x_1, \ldots, x_n$ in $\mathbb{R}^d$ (we will later normalize them). Next, we set $q_i = \frac{1}{\sqrt{|S_i|}} \sum_{j \in S_i} x_j$. Note that, by stability of Gaussian random variables and independence of the $x_j$'s, it follows that each coordinate of $q_i$ is distributed independently as a standard normal distributed (i.e. $\mathcal{N}(0, 1)$). Thus, by standard $\chi^2$ concentration (e.g. Lemma 1 Laurent & Massart (2000)), for any fixed $i \in [n]$ and value $\lambda > 0$, if $g \sim \mathcal{N}(0, I_d)$ is a vector of i.i.d. standard Gaussian variables, we have $\|g\|_2^2 = d + 2\sqrt{d\lambda} + 2\lambda$ with probability at least $1 - 2^{-\lambda}$. Setting $\lambda = 4 \log n$ and taking $d$ larger than some constant, we have

$$\mathbf{Pr}\left[ \left| \|g\|_2^2 - d \right| \leq 5\sqrt{d \log n} \right] \geq 1 - n^{-4}$$

We can thus condition on $\|x_i\|_2^2 = d \pm 5\sqrt{d \log n}$ and $\|q_i\|_2^2 = d \pm 5\sqrt{d \log n}$ occurring for all $i \in [n]$, which holds with probability $1 - 2n^{-3}$ by a union bound over the $2n$ vectors. Call this event $\mathcal{E}_1$.

To analyze this construction, first define the event $\mathcal{E}_2$ that for all $(i, j) \notin E$, we have $|\langle q_i, x_j \rangle| \leq 100\sqrt{d \log n}$. Also define on the event $\mathcal{E}_3$ that for all $(i, j) \in E$ we have $|\langle \sum_{t \in S_i \setminus j} x_t, x_j \rangle| < 100\sqrt{ds \log n}$.

We first analyze $\mathbf{Pr}[\mathcal{E}_2]$. If $(i, j) \notin E$, then $q_i, x_j$ are independent Gaussian vectors, thus by Gaussian stability we have

$$|\langle q_i, x_j \rangle| \sim |g| \cdot \|x_j\|_2 \leq |g|\sqrt{d}\left( 1 + 5\sqrt{\frac{\log(n)}{d}} \right)^{1/2} \leq |g|\sqrt{d}(1 \pm 1/100)$$

where $g \sim \mathcal{N}(0, 1)$ and we took $d$ larger than some constant. Via the density function of a Gaussian, we have $\mathbf{Pr}\left[ |g| \cdot \|x_j\|_2 > 100\sqrt{d \log n} \right] < 1/n^4$. Thus, by a union bound over at most $n^2$ pairs, we have $\mathbf{Pr}[\mathcal{E}_2] > 1/n^2$. For $\mathcal{E}_3$, note that for any $i \in [n]$ with $j \in S_i$, the vector $\sum_{t \in S_i \setminus j} x_t$ is distributed like $\mathcal{N}(0, \sqrt{|S_i| - 1} \cdot I_d)$; namely each coordinate is i.i.d. Gaussian distributed with variance $|S_i| - 1$. Thus

$$\left| \left\langle \sum_{t \in S_i \setminus j} x_t, x_j \right\rangle \right| \sim \sqrt{|S_i| - 1} \cdot |g| \cdot \|x_j\|_2 < |g|\sqrt{sd}(1 + 1/100)$$

where again $g \sim \mathcal{N}(0, 1)$. Following the same argument as above yields $\mathbf{Pr}[\mathcal{E}_3] > 1 - n^{-2}$.

Conditioned on $\mathcal{E}_1, \mathcal{E}_2, \mathcal{E}_3$, we claim that the vectors $q_1/\|q_1\|_2, \ldots, q_n/\|q_n\|_2, x_1/\|x_1\|_2, \ldots, x_n/\|x_n\|_2$ satisfy the desired properties with $r = \frac{1}{4\sqrt{\lambda}}$. In what follows, let $\lambda = 10 \cdot \max\{s, \frac{1}{\epsilon^2}\}$. For case one, if $j \in S_i$ we have

$$\left\langle \frac{q_i}{\|q_i\|_2}, \frac{x_j}{\|x_j\|_2} \right\rangle = \frac{1}{\|q_i\|_2 \|x_j\|_2 \sqrt{|S_i|}} \left( \|x_j\|_2^2 + \left\langle \sum_{t \in S_i \setminus j} x_t, x_j \right\rangle \right)$$

$$\geq \frac{2}{3d\sqrt{\lambda}} \left( \frac{2}{3}d - 100\sqrt{ds \log n} \right) > \frac{4}{9\sqrt{\lambda}} - \frac{200}{3} \cdot \sqrt{\frac{\log n}{d}} \qquad (3)$$

$$\geq \frac{1}{3\sqrt{\lambda}}$$

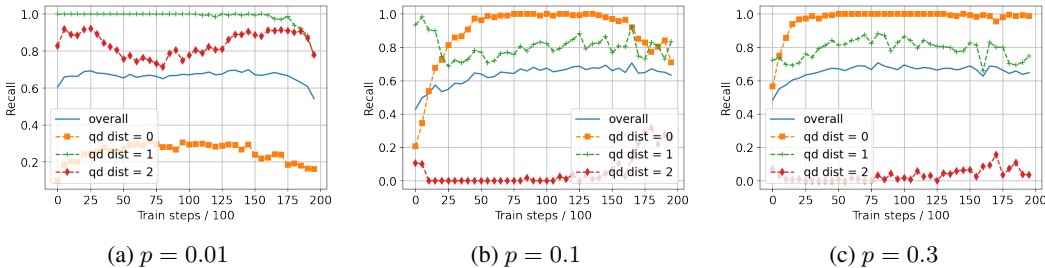

(a) $p = 0.01$     (b) $p = 0.1$     (c) $p = 0.3$

Figure 6: Effect of re-balanced training data with a varying ratio $p$ between the regular and the heavy-tail data on retrieval quality.

Where we used that $d \geq C\lambda \log n$ for a sufficiently large constant $C$. Next, for case two, when $j \notin S_i$, we have

$$\left\langle \frac{q_i}{\|q_i\|_2}, \frac{x_j}{\|x_j\|_2} \right\rangle < \frac{3}{2d} \left\langle \sum_{t \in S_i \setminus j} x_t, x_j \right\rangle \leq \frac{3}{2\sqrt{d}} \cdot 100\sqrt{\log n} \leq \frac{1}{5\sqrt{\lambda}} \tag{4}$$

Using that $d \geq C \max\{1/\epsilon^2, s\} \log n$, we have that $\frac{1}{5\sqrt{\lambda}} + \epsilon < \frac{1}{4\sqrt{\lambda}} < \frac{1}{3\sqrt{\lambda}} - \epsilon$, which completes the proof.

$\square$

# B   ADDITIONAL EXPERIMENTS

In Figure 6, we provide additional results complementing Figure 3b in showing that rebalanced sampling cannot effectively solve the lost in the long distance problem.

