# OpenReview forum: "Asymmetric Embedding Models for Hierarchical Retrieval: Provable Constructions and a Pretrain-Finetune Recipe"
_ICLR.cc/2025/Conference — Submitted to ICLR 2025_

### Official Review · Reviewer_K7bM · 2024-10-27

**Soundness:** 2
**Presentation:** 2
**Contribution:** 2
**Rating:** 3
**Confidence:** 3

**Summary:**

This paper investigates the problem of hierarchical retrieval with dual encoders. The authors first provide a theoretical analysis of the hierarchical retrieval problem, proposing a constructive algorithm (algorithm 1) and demonstrating the dimension of the embeddings needed under a specific boundary condition. Then, they learn a dual encoder in a perfect W-trees hierarchical retrieval setting and show that the dimension for such a model can be smaller than the handcrafted embedding proposed in the previous section. After that, the author identifies the lost in the long-distance problem and proposes a pretrain-finetune recipe. Finally, they conduct experiments on WordNet to demonstrate the effectiveness of their proposed method.

**Strengths:**

- The paper researches an interesting problem of hierarchical retrieval under the dual encoder setting.
- The paper provides theoretical analysis and empirical experiments on the proposed method.

**Weaknesses:**

- The paper is not well written and some claims are not demonstrated strictly: In section 3, the author mentions that the goal is to find embeddings for queries and documents that satisfy Cases 1 and 2. Then, they propose a constructive algorithm in “Algorithm 1”. However, there is no demonstration that “Algorithm 1” satisfies the Cases 1 and 2.

- The proposed scenarios are too ideal and may not aligned with real-world scenarios. In the real world, queries and documents are usually associated with some textual information which is very important for semantic search. However, in this paper, the authors ignore such information and assume just a pure embedding for each query/document (if semantic is considered, some text encoder should be introduced).

- It seems that most of the experiments are done on synthetic datasets. Even though there is a final section on WordNet, I would like to challenge that the retrieval problem on such a dataset is also artificial. Given that the author provides an example of ads search or product search with hierarchical retrieval in the introduction, I would like to recommend the author to conduct experiments on such datasets.

- Some experimental settings are not clear and reasonable to me. Why do the authors assume that the query is also in the hierarchical tree in Section 4? What are the documents that are relevant to q on the tree? It is not clear how the training and evaluation set up is.

- The perfect W-tree assumption is very strong. In the real world, the hierarchy is not guaranteed to be balanced. I would challenge if the findings in the paper can be generalized to real-world scenarios.

- Algorithm 1 seems to be very naive and the motivation of providing the study in Figure 2 is not clear. It seems that such a method has a very poor performance compared to the learned embedding. Even for the learned embedding method, since the authors do no introduce a text encoder, it does not reach its maximum capability.

- Why hyperbolic embeddings cannot be adopted in neighbor search algorithms? Can a similar pretrain-finetune strategy be adopted for hyperbolic embedding?

- The paper would be improved by adding a related work section.

- Since the paper only researches the pure embedding cases, I would challenge the application of such a method in 2024 given that we can usually use more advanced text encoder for representation or use LLMs for search.

**Questions:**

See the weakness section.

---

### Official Review · Reviewer_Ky5E · 2024-11-03

**Soundness:** 2
**Presentation:** 2
**Contribution:** 2
**Rating:** 5
**Confidence:** 4

**Summary:**

For a retrieval task, an encoder is used to embed the query and the document, and then the most similar documents are retrieved by similarity matching. In the context of hierarchical retrieval, this type of modeling ignores the Euclidean geometry of the embedding space, neglecting the conceptual hierarchy of documents and thus retrieving irrelevant documents. The paper contributes to hierarchical information retrieval using Dual Encoders (DEs) by developing a novel constructive algorithm that generates embeddings for hierarchically structured data.
Embeddings are constructed by assigning random Gaussian vectors to document nodes and creating query embeddings as normalized sums of relevant document vectors, enabling retrieval by proximity in Euclidean space. This approach ensures that hierarchical relationships are represented, allowing relevant ancestors to be retrieved for each query. A synthetic dataset is constructed on a W-tree to sample training and evaluation sets for conducting experiments. A key limitation of the strategy, where DEs struggle to retrieve documents far from the query, is identified and mitigated by a pretrain-finetune strategy. The real-world experiments are performed on WordNet.

**Strengths:**

The paper provides a constructive algorithm with theoretical proofs to mitigate the problem of document retrieval. The approach has been validated on carefully constructed dataset, identifying the limitation and finally proposing a solution to the problem.

**Weaknesses:**

1. The real-world experiments are limited to a constrained vocabulary dataset from WordNet. It would strengthen the paper to include more real-world experiments on datasets such as ConceptNet, which combines hierarchical and non-hierarchical relationships. Additionally, structured datasets like the Microsoft Academic Graph, which includes both topic and citation hierarchies, would provide valuable insights into the approach's effectiveness in diverse hierarchical structures.
2. My high level comment is the paper has a very limited scope unless presented with more real world experiments.
3. The idea of hyperbolic and Euclidean space has been explored but not fully explored in the paper. Real-world datasets often do not possess such straight forward relationships thus would be good to see this section in more detail outlining the comparison and pitfalls of both approaches.

**Questions:**

Were there any experiments conducted with varying query and document lengths? If not it would be good to add a small paragraph on varying embedding sizes.

---

### Official Review · Reviewer_eRur · 2024-11-03

**Soundness:** 2
**Presentation:** 3
**Contribution:** 2
**Rating:** 5
**Confidence:** 4

**Summary:**

Motivated by key-word matching in advertisement the paper addresses the problem of hierarchical retrieval, meaning, the retrieval problem where if a document is relevant to a given query then its ancestors will also be relevant. In this context the paper is set to address the limitations of embeddings spaces to represent geometrical structures. The paper first provides scaling rule for the (upper bound of) number of dimensions required to achieve satisfactory retrieval performance. As part of this work the authors propose an algorithm to construct such embeddings spaces. They also compare the performance of learned embeddings with this hand-crafted approach and see that learned embeddings require significantly less dimensions. Last, the authors present a continual pre-training approach (referred to as pretraining-finetuning) to address the long tail problem.

**Strengths:**

The paper is well structured and focuses on analyzing and understanding the limitations of popular retrieval approaches in the ambit of hierarchical structures. It provides insightful results about a novel application of continuous pertaining, commonly used to address the long-tail distribution problems of long documents in LLMs, can also be employed to address the "long distance ancestor-retrieval" in hierarchical retrieval and show that it outperforms standard up-sampling approaches.

**Weaknesses:**

1. It is unclear what is the benefit of the handcrafted approach in section 3 as the learned embeddings require significantly less dimensions to achieve the same level of performance.

2. The theoretical upper boundary is a rather conservative one as learning embeddings can achieve the same performance at orders of magnitude less dimensions.

3. While the learning embeddings is a sound baseline the paper (direct application of InfoNCE to learn the hierarchical embeddings), the paper does not provide previous reference about current state of the art to address this problem. It is unclear from the paper if the current practice is to use non-euclidean distance that are not supported in DBs, naive euclidean distances, etc.

4. During the constructions of the training and evaluation sets it is unclear the overlap between training and evaluation. For example based on the described process in section 4 and given a tree path (a_1, a_2, a_3) where a_i is the parent of a_i-1 for i=2,3. the pair (a_1, a_2) can belong to the training set and the pair (a_2, a_3) to validation leading to leakage. Moreover, the construction does not account for grammatically different and semantically similar queries implying in section 4 and 5 that the queries has to be part of the tree.

5. The method proposed in Section 5 is a direct application of continuous pre-training. however, the paper does not provide any reference to this work.

**Questions:**

I think the main contribution of the paper is the application of continuous learning to address the "long distance ancestor-retrieval" problem. Thus, the paper could benefit from a re-writing around this contributions and a more thorough experimentation and benchmarking of the same against the existing approaches to address the problem (not only in non-euclidean geometries but also in euclidean geometries) alongside a more thorough background section articulating (1) what are the current approaches used in conjunction with euclidean distance and how the proposed methods compare to these and (2) a literature summary on continuous pre-training.

---

### Official Review · Reviewer_6jzd · 2024-11-04

**Soundness:** 2
**Presentation:** 2
**Contribution:** 1
**Rating:** 1
**Confidence:** 4

**Summary:**

This Information Retrieval paper tackles the problem of training a dual encoder in the case where the document collection is represented as a directed acyclic graph. More specifically, the authors suppose that if one node is relevant, then all its ancestors are relevant. This is motivated by keyword targetting search, e.g. "Kid's running shoes" should retrieve "Kid's running shoes", "Kid's shoes" and "Running shoes".

The authors first show (section 3, theorem 1) that given a set of queries and documents, it is always possible to construct embeddings (algorithm 1) so that a query $q$ will have $q \cdot d \ge r+\epsilon$ for any relevant document $d$ (and $\ge r-\epsilon$ for not relevant ones). The authors then show experimentally (section 4) that learning the embeddings is much more effective than constructing them with the algorithm 1.

Section 5 deals with improving hierarchical retrieval - the main propositions are (1) to oversample relevant documents which are more distant than the most specific ones; (2) to pre-train on balanced data and then fine-tune with oversampled relevant documents. Experiments are conducted in section 6 on WordNet (and Hyperlex) showing that this method improves over a simple training procedure.

**Strengths:**

The paper proposes a new ranking problem with target "documents" linked within a directed acyclic graph. The proposed modification to a standard training of a dual encoder are simple.

**Weaknesses:**

There are several weaknesses in this paper.

First, the overall motivation is not clear. If a DAG for documents exists, then it could be leveraged directly by retrieving only the most specific nodes and then using the hierarchy to retrieve their ancestors. It could be argued that this hierarchy is only implicit and has to be discovered during training, but in that case, the proposed training procedure could not be used, or would probably lose any interest when generalizing to new documents. The authors should thus clarify the motivation for using a single space to represent all the matching documents. I would also suggest looking at Matryoshka embeddings that could be an inspiration when dealing with such DAGs.

Second, it is surprising to not modify equation 1 (l. 145) in the case of multiple relevant documents - this would allow to include nodes that are more or less distant to the most specific one(s), potentially avoiding the problems discussed in section 4 without using a complex training procedure. Especially, since the batch size used is quite high (4096) given the number of "documents" in WordNet (82,115 documents), it is quite likely to sample false negatives when computing eq. (1). Modifying equation 1 to include more relevant documents for a query, and/or ensuring that no false negatives are introduced into the equation, would be some possible simple solutions.

Third, algorithm 1, which is used when demonstrating theorem 1, cannot always separate the embeddings as specified (i.e. with a margin $\epsilon$) - this will only be achieved with some probability (which are linked to the events $\mathcal E_1$, $\mathcal E_2$ and $\mathcal E_3$). Thus it would have been more interesting to look at the separation probability of such an algorithm. Also, the proof in the appendix is hard to read, as notations and arguments could help the reader to understand better the different steps (I am still unsure of many parts in this demonstration).

Fourth, The experiments are only conducted on very small-scale datasets (82K documents for WordNet). How are the datasets representative of the use case (i.e. keyword matching)? Note that classification datasets such as the "PubMed MultiLabel Text Classification Dataset" could be used in a retrieval setting.

Fifth, there are no related works sections - that should cover IR models, and also hierarchical classification/retrieval works.


The paper also could be better written (there are also a few typos):
- l. 84-91: hyperbolic embeddings could be discussed there rather than at the end of the paper, since they were designed to capture hierarchies
- l. 100-102 / "long(er) distances" is not defined
- l. 162: when minimizing
- l. 183: $s$ is not defined
- proof of theorem 1 (Appendix A.1.) is quite unclear (see questions)
- l. 561-562: notation $d \pm ...$ is not precise (this does not denote an interval as intended).
- l. 466, there are no related works

**Questions:**

1. The cases l. 187-189 suppose that the model is well calibrated - which is a very strong hypothesis. Could you justify it?
1. In proof of theorem 1, I don't get how you prove eq. at line 572 since the distribution of $q_i \cdot x_j$ is not a chi-squared. Also, the $\sim$ symbol suggests that they follow the same probability distribution, but then what does it mean for a probability distribution to be bounded?

---

### Meta-Review · Area_Chair_y44z · 2024-12-16

**Metareview:**

This paper considers the problem of hierarchical retrieval as instantiated, e.g., in keyword search. There was consensus amongst reviewers that the technical motivation here is not adequately articulated (see especially comments from reviewer 6jzd), and the experimental setup was also seen as quite weak. The authors did not respond to these concerns raised in review.

**Additional Comments On Reviewer Discussion:**

The authors did not offer a response to the reviews.

---

### Decision · Program_Chairs · 2025-01-22

Reject